# Preference for Animals: A Comparison of First-Time and Repeat Visitors

**Yulei Guo [1,\*] and David Fennell [2]**

[1] Chengdu Research Base of Giant Panda Breeding, Sichuan, Chengdu 610081, China
[2] Department of Geography & Tourism Studies, Brock University, St. Catharines, ON L2S 3A1, Canada; dfennell@brocku.ca
\* Correspondence: yulei.guo@panda.org.cn

**Abstract:** Wildlife tourism is one of the strongest-performing sectors in the global tourism market. While tourists' preferences for and affection towards animals are a cornerstone of the industry, a better understanding of how experiences, including animal–tourist encounters and visitation frequency, influence visitors' animal preferences is required. Through a comparison of preferences among first-time and repeat visitors of four species (giant panda "*Ailuropoda melanoleuca*", red panda "*Ailurus fulgens*", peafowl "*Pavo cristatus*", and swan "*Cygnus*"), both before and after animal encounters at the Chengdu Research Base of Giant Panda Breeding (Panda Base), the results show that different species elicit varied and, at times, contrasting tourist preferences. As a result, animal preferences in wildlife tourism can vary based on different stages of visitation. Highlighting this dynamic relationship between animal preferences and visitation experiences is further elucidated through consumer learning theory and lively capital. The outcomes of this study contribute to a deeper grasp of human–animal interactions and have broader implications for the development of conservation programs in captive wildlife venues.

**Keywords:** preference for animals; first-time visitors; repeat visitors; animal encounters; consumer learning; lively capital; animal conservation





## 1. Introduction

Wildlife tourism is one of the strongest-performing sectors in the global tourism market. In the first decade of the new millennium, statistics indicate that approximately 12 million wildlife tourism trips were taken, with an annual growth rate of 10% per annum [1]. In 2021, the wildlife tourism market's estimated base was USD 28 billion, growing to USD 135 billion in 2022 and forecasted to be USD 219.9 billion by 2032 [2]. In Africa, over one-third of all tourism revenue was attributed to wildlife tourism in 2018 [3], a statistic representative of wildlife tourism in many other country contexts [4].

The sheer volume of tourists at wildlife tourism attractions (WTAs) indicates the affection people show towards animals [5,6]. The subconscious love for animals is explained in Wilson's [7] concept of biophilia, defined as humans' innate emotional affiliation to other living organisms [7]. This has also been investigated by Kellert [8], based on ten primary attitudes that Americans hold toward wildlife. Kellert [8] found that the most prevalent attitudes are humanistic, moralistic, utilitarian, and negativistic, with two primary attitudinal clashes along utilitarian–moralistic and negativistic–humanistic lines. Preferred animals, Kellert [9] found, were those that were larger, attractive (aesthetically), thought to be intelligent, and which have a history of association or benefit with/to humans. People's perceptions [10–14] of animal traits [15–21] can determine visitors' preferences and affection towards animals, which in turn influence programs and projects in animal welfare [22–26] and conservation education [27–29].

However, more recent studies have challenged the perception that emotion and affections are merely "private matters" belonging to individuals. Ahmed [30] (p. 117) suggests

that emotions circulate between bodies and "create the very effect of the surfaces or boundaries of bodies and worlds". Feelings and affection towards animals are not the end-product of animal–visitor encounters but rather what circulates and exchanges between bodies. Moreover, to prefer and to be affected involves movements and active associations with certain "feelings". In a different yet highly compatible context, Haraway [31] introduced the notion of encounter value between species and lively capital as a product of these encounters. Encounter value builds on the intersection between bodies where emotions, affection, and preference move to materialize surfaces and borders. Lively capital suggests that this exchange between bodies based on emotions and affections is interwoven with capitalist society. Hence, for both Ahmed and Haraway, the affection and preference towards animals is not a readily existing attitude/perception to be commercialized. Instead, animal affection and preference emerge as a process resulting from the interspecies encounter, while lively capital results in value accumulation or loss.

This study investigates the generation of encounter value and lively capital in wildlife tourism stemming from two main questions:

1. Does the animal–visitor encounter contribute to changes in animal preference and affection?
2. Do frequent animal–visitor encounters result in greater affection and preference towards animals?

The first question probes the creation of encounter value in wildlife tourism. Instead of asking how tourists' affection and preference toward animals influence their visitation experience, this paper asks what this animal encountering experience means for visitors. Question 2 extends our understanding of encounter value and lively capital by investigating the relationship between visiting frequency, affection, and animal preference.

One implication of applying the notion of lively capital to wildlife tourism is that the animal–visitor encounter is a dynamic and interactive value creation process. In order to capture the nature of this ongoing and interchanging process, this paper employs consumer learning as the primary theoretical construct.

## 2. Literature Review

Defined as "the process by which individuals acquire the purchase and consumption knowledge and experience that they apply to future related behaviour" [32] (p. 166), consumer learning positions lively capital and encounter value in a continuous dialogue. According to Schiffman et al. [32], learning is a process; it evolves as a function of new knowledge or through experience, which provides feedback for future related behaviour. Contextualized in consumer learning, visitors' affection and preference towards animals is a process of gaining new insights through animal–visitor encounters, influencing animal affection and preference.

Theorists [33–35] argue that there are two general learning theories: behavioral and cognitive. Regarding the behavioral theory, Niosi [36] believes that learning responds to external events, where the feedback people receive from events and circumstances influences their experiences over time. The World Wildlife Fund's use of the giant panda (Ailuropoda melanoleuca) to promote conservation and wildlife protection is a case in point [37]. In contrast, cognitive learning theories focus on "internal mental processes where people are viewed as problem solvers using information obtained from the world to master the environment" [36] (p. 57). There are two main cognitive learning theories. The first is observational learning, whereby individuals alter their attitudes and behaviors by watching others, and where there is what Niosi [36] refers to as vicarious rather than direct experience. These experiences are stored in memory and used at a later point in time. The second cognitive learning theory is modelling, where the individual imitates the actions of others, with these actions used in the future. For example, if tourists are mindful of the welfare and conservation issues confronted by animals drawn into the industry, the individual may replicate or model these behaviors in the future.

Studies focusing on wildlife tourism have often considered the animal–visitor encounter as an external event modifying visitors' behaviour. For example, In China, consumptive and non-consumptive (captive, semi-captive, and wild settings) wildlife tourism has advanced considerably [38]. Studies of human–giant panda interactions show that the motivation to be close to pandas and even sit with them or hold them (for a hefty price) increases satisfaction [39]. Research also shows differences between international and domestic tourists in terms of wanting to see giant pandas in captive and wild settings, with the former more motivated to see pandas in a captive environment than the latter [40]. In a study of wildlife tourists to the Giant Panda National Park and Breeding Centre, wildlife tourists not strongly motivated to engage in wildlife observation only focused on the Giant Panda Breeding Center rather than exploring further into the park, suggesting the need to keep tourists longer and to expose them to more education [41]. Intensive and prolonged education programs have positively changed knowledge and attitudes toward pandas by enhancing compassion and empathy [42].

Animal–visitor encounters can also serve as cognitive learning experiences through which tourists acquire information to build relationships with the environment. This branch of studies has focused on the connection between learning, place attachment, and loyalty. For example, Cui et al. [43] found that tourists to Panda Base who were satisfied with interpretation programs were more likely to enhance nature and place attachment, which further enhanced Panda Base loyalty. Consumers' increased destination loyalty led to stronger belief in the value of ecotourism and the amount of money they were willing to pay at the Panda Base [43]. Research also shows that higher-status animals, like giant pandas, are held in higher moral regard than other lower-status animals, like frogs and bats [44], with pandas given the status of cuddly wild animals under the broader category of "Level of Charisma" [45]. This high level of charisma (the panda Chi Chi at the London Zoo inspired the WWF logo) has catalyzed the use of pandas as ambassadors at zoos around the world, increasing attendance [46] and significantly enhancing customer loyalty [47].

Despite the recognition that tourists' biophilic interest in animals drives the growth of the wildlife tourism industry, existing studies still need to address the relationship between the visitor experience and affection and preference toward animals. This study empirically investigates the wildlife tourism experience as a learning process tied to tourists' affection and preference for animals. We use two key tourism experience measures—visitation frequency and animal encounters—as critical turning points for visitors' interest in animals. In recognition of the contextual bond between animals and humans [14], our study is anchored on the fluctuation of tourist preferences for the four animals (giant pandas, red panda "*Ailurus fulgens*", peafowl "*Pavo cristatus*", and swan "*Cygnus*") at the Panda Base and by asking whether participants had good interactions with these four animals. Building on relevant studies [14,48–55], we view visitors' affection and preference for animals as a multidimensional construct that includes ten aspects (Table 1), explained in more detail below.

**Table 1.** Factors influencing the public's preference for four animals at the Panda Base.

| No. | Factors |
| --- | --- |
| 1 | Aesthetics: Animals considered cute are more attractive and, therefore, more preferred. |
| 2 | Aggressive behaviour: Animals that pose a perceived threat or danger to people are less liked. |
| 3 | Intelligence: Animals considered to possess the capacity for reason, feeling, and emotion are preferred. |
| 4 | Human–animal interaction: Animals interacting more with people are preferred. |
| 5 | Degree of freedom: Animals enjoying greater freedom are preferred. |
| 6 | Traditional culture of China: Animals that play an essential role in the history or culture of China are likely to be preferred. |
| 7 | Pet ownership: Animals that are pets or useful to humans will likely be preferred. |
| 8 | Willingness to donate: Preferred animals are more likely to attract donations. |
| 9 | Willingness to know: The public wants to learn more about the preferred animals. |
| 10 | Existing knowledge about the animal: The public has a deeper understanding of preferred animals. |

### 3. Research Method

*Data Summary and Research Background*

Located in Chengdu, China, the Panda Base has bred the world's largest population of captive giant pandas since 1987. As a result of its pioneering efforts in wildlife protection and biodiversity conservation, the Panda Base has become Chengdu's most visited tourist attraction, drawing in as many as 9 million tourists in 2019 [56]. The Panda Base displays giant pandas across different lifespans and several companion species, including peafowls, red pandas, and swans.

The research team collected a gender-, age-, origin-, and education-defined sample of tourists visiting the Panda Base in July and October 2022 via the survey platform Wenjuanxing ("问卷星" "Survey Star"). Participants accessed the questionnaire via a QR code issued by Wenjuanxing, which the research team printed on a poster. The anonymous survey ensured that no risk would come to the participants, who could withdraw from the survey at any moment. All participants offered informed consent when accessing the questionnaire. As children were allowed to participate in the study and are a primary visitor cohort to the Panda Base, the researchers allowed parents to assist children in completing the questionnaire. Chinese law allows anonymous studies, such as this one, which do not utilize personally identifiable information, such as biometrics, religion, medical health, financial account, or location, without the approval of an ethical committee.

Table 2 illustrates the visitor groups included in this study. Two critical moments of the visiting experiences—the visitation frequency and animal encounters—offered boundaries for dividing the sample. Visitors at the entrance and exit of Panda Base volunteered to participate in the questionnaire. The research team offered a giant panda magazine for participants as a gift for participation in the study. Because one of the authors is an employee at the Panda Base and has conducted extensive fieldwork on visitors, the researcher knows that visitors at the exit were more likely to identify with Group 1. Group 2 data were collected at the entrance at the Panda Base.

**Table 2.** Grouping participants depending on visiting frequency and animal encounters.

|  | Have Encountered Animals | Have Not Encountered Animals |  |
| --- | --- | --- | --- |
| First-time visitor | Group 1a (168) | Group 2a (163) | Group a (331) |
| Repeat visitor | Group 1b (169) | Group 2b (159) | Group b (329) |
|  | Group 1 (337) | Group 2 (322) | 659 |

Additionally, a question in the survey asked participants to report whether they had encountered animals at the Panda Base in order to establish two sample groups. In total, 667 tourists participated in the survey; however, 5 participants failed to answer all questions, and 3 offered unrealistic answers for their ages. Therefore, these 8 participants were excluded, leaving a final sample of 659.

Over 80% of Panda Base tourists were first-timers, making them easier to survey than repeat visitors. Thus, extra effort was employed to target repeat visitors, which was made explicit on survey posters. The online questionnaire filtered respondents via two initial questions: (1) Is this your first Panda Base visit? (2) Have you seen pandas already? Based on these answers, participants either proceeded or were screened out. The team monitored the number of participants in each group. Collecting sufficient data for Group 2B took the longest time.

Ten dimensions assessed affection and preference for four animals that tourists encountered or would encounter at the Panda Base (Table 1). A questionnaire was designed based on an interpretation of the two dimensions (see Supplementary Material). At the Panda Base, tourists were exposed to educational and interpretive messages, which allowed them to increase their knowledge of the four animals included in the study. However, approximately 90% of these messages targeted the giant panda, with only 5–10% targeting

the red panda. There is no mention of peafowls and swans in any educational or interpretive format. The ten measures used were modified from the factors which influence preference for animals [14], where respondents are asked to evaluate their affection and preference towards the four animals. For example, for the cuteness dimension, the animal selected first was the animal deemed cutest, and the animal placed last was the animal deemed to be least cute. All participants evaluated the four animals on the ten dimensions, which assessed the public's preference for these animals. In parallel to the ten dimensions, participants also directly evaluated their preference and affection for the four animals.

Since four animals were included in this study, a four-point differential scale was employed to assess each dimension. For example, in the cuteness dimension, the cuter the respondent perceived the animal, the lower the score (Table 3). According to Hout, Papesh, and Goldinger [57], multidimensional scaling (MDS) provides a tool to quantify similarity judgments and reveals relational structures among evaluated items. For this study, MDS is highly suitable for observing how animal encounter experiences create lively capital that changes the affection and preference towards animals.

**Table 3.** Multidimensional animal preference scale used in the study.

| Preference | Scale | Preference | Scale |
|---|---|---|---|
| Cute | 1 | Not cute | 4 |
| Aggressive | 1 | Unaggressive | 4 |
| Intelligent | 1 | Unintelligent | 4 |
| Interactive | 1 | Inaccessible | 4 |
| Greater freedom | 1 | Restricted | 4 |
| Means a lot in traditional Chinese culture | 1 | Does not | 4 |
| Represents a national image | 1 | Does not | 4 |
| Pet ownership | 1 | Not a pet | 4 |
| Willingness to donate | 1 | Not willing | 4 |
| Willingness to know | 1 | Not willing | 4 |
| Has existing knowledge about the animal | 1 | Does not | 4 |

Skewness and kurtosis tests were run to test if the data were normally distributed. The acceptable range for skewness falls between −2 and +2, and kurtosis is appropriate based on a range of −7 to +7 [58,59]. Table 4 shows that data relating to giant pandas was not normally distributed. Also, some datasets concerning red pandas, peafowl, and swans were also not normally distributed. The main reason for this is that the histogram is highly skewed given the prominence (high ranking) of the giant panda compared to the other three animals.

ANOVA and Kruskal–Wallis tests were conducted in SPSS to provide further insight into the samples. ANOVA assumes that the data have a normal distribution, while the Kruskal–Wallis test does not assume normality. Both tests compared differences between the four visitor groups, assuming that generating lively capital during the visits could shape visitors' preferences and affections toward the animals.

**Table 4.** Normality of the samples.

| Measures | Group 1a Skewness | Group 1a Kurtosis | Group 2a Skewness | Group 2a Kurtosis | Group 1b Skewness | Group 1b Kurtosis | Group 2b Skewness | Group 2b Kurtosis | ANOVA or Kruskal–Wallis * |
|---|---|---|---|---|---|---|---|---|---|
| | | | | GIANT PANDA | | | | | |
| Cuteness | 3.219 | 12.798 | 5.135 | 28.165 | 3.703 | 16.503 | 4.154 | 15.447 | K |
| Unaggressiveness | 1.045 | −0.426 | 0.642 | −1.207 | 1.315 | 0.248 | 0.782 | −0.981 | A |
| Intelligence | 2.017 | 2.94 | 2.105 | 3.288 | 2.095 | 3.532 | 2.426 | 5.039 | K |
| Interaction with tourists | 1.71 | 1.657 | 2.115 | 3.411 | 1.676 | 1.477 | 2.644 | 6.126 | K |
| A free lifestyle | 1.285 | 0.022 | 1.561 | 1.02 | 1.42 | 0.41 | 1.927 | 2.063 | A |
| Chinese traditional culture | 4.209 | 17.441 | 6.153 | 43.564 | 5.573 | 34.617 | 6.317 | 42.819 | K |
| Pet ownership | 3.091 | 9.195 | 2.32 | 5.141 | 3.942 | 17.614 | 2.961 | 8.72 | K |
| Donation | 4.034 | 18.161 | 4.524 | 19.513 | 4.059 | 17.234 | 4.696 | 26.698 | K |
| Willingness to learn | 3.768 | 15.694 | 5.1 | 28.16 | 3.485 | 12.325 | 4.911 | 27.347 | K |
| Knowledge about the animal | 3.647 | 14.012 | 5.164 | 27.794 | 4.164 | 18.034 | 4.976 | 25.229 | K |
| Preference | 3.176 | 10.721 | 4.748 | 24.573 | 3.103 | 9.587 | 4.837 | 23.91 | K |
| | | | | RED PANDA | | | | | |
| Cuteness | 1.118 | 2.148 | 1.053 | 0.051 | 1.151 | 3.371 | 1.769 | 0.35 | A |
| Unaggressiveness | 0.664 | 0.124 | 0.626 | −0.331 | 0.713 | 0.33 | 0.753 | 0.593 | A |
| Intelligence | 0.842 | 1.844 | 0.568 | 1.334 | 0.489 | 1.428 | 1.272 | 3.715 | A |
| Interaction with tourists | 0.9 | 1.252 | 1.004 | 1.314 | 0.806 | 0.786 | 0.912 | 2.684 | A |
| A free lifestyle | 0.678 | 0.763 | 0.831 | 0.677 | 0.784 | 1.489 | 1.433 | 3.326 | A |
| Chinese traditional culture | 1.473 | 1.801 | 0.018 | 4.523 | 1.48 | 0.856 | 1.878 | 3.025 | A |
| Pet ownership | 1.097 | 4.775 | 0.042 | 7.182 | 1.234 | 3.137 | 0.785 | 3.724 | K |
| Donation | 1.189 | 4.486 | −0.178 | 10.758 | 1.666 | 5.155 | 1.635 | 6.672 | K |
| Willingness to learn | 1.304 | 5.747 | −0.332 | 12.712 | 1.543 | 5.08 | 1.887 | 8.471 | K |
| Knowledge about the animal | 1.374 | 5.545 | −0.247 | 12.126 | 1.665 | 3.819 | 2.103 | 5.085 | K |
| Preference | 1.352 | 5.172 | −0.525 | 13.662 | 1.529 | 4.171 | 1.873 | 10.669 | K |
| | | | | PEAFOWL | | | | | |
| Cuteness | −0.611 | 1.419 | −0.35 | 0.134 | −0.385 | 1.466 | −0.325 | 0.411 | A |
| Unaggressiveness | −0.579 | 0.291 | −0.497 | −0.149 | −0.741 | −0.021 | −0.606 | −0.503 | A |
| Intelligence | −0.812 | 1.03 | −0.585 | 1.417 | −1.03 | 2.038 | −1.024 | 2.77 | A |
| Interaction with tourists | −1.003 | 0.881 | −1.058 | 1.47 | −0.81 | 0.219 | −0.831 | 1.947 | A |
| A free lifestyle | −1.012 | 1.081 | −0.76 | 1.021 | −0.922 | 0.485 | −1.009 | 1.647 | A |
| Chinese traditional culture | −0.608 | 1.107 | −2.747 | 18.382 | −0.725 | 1.696 | −0.245 | 1.441 | K |
| Pet ownership | −0.741 | 4.069 | −2.975 | 24.251 | −0.634 | 2.925 | −0.974 | 2.937 | K |
| Donation * | −0.396 | 3.834 | −3.502 | 29.338 | −0.534 | 3.836 | 0.45 | 1.094 | K |
| Willingness to learn * | −0.858 | 5.014 | −4.011 | 36.629 | −0.443 | 5.891 | −0.627 | 3.73 | K |
| Knowledge about the animal | −1.028 | 4.915 | −4.16 | 36.698 | −1.015 | 4.282 | −0.476 | 2.667 | K |
| Preference | −0.712 | 4.256 | −3.685 | 29.728 | −0.535 | 4.408 | −0.08 | 3.3 | K |

**Table 4.** *Cont.*

| Measures | Group 1a | | Group 2a | | Group 1b | | Group 2b | | ANOVA or Kruskal–Wallis * |
|---|---|---|---|---|---|---|---|---|---|
| | Skewness | Kurtosis | Skewness | Kurtosis | Skewness | Kurtosis | Skewness | Kurtosis | |
| | | | | SWAN | | | | | |
| Cuteness * | −1.831 | 3.594 | −1.023 | 0.03 | −1.726 | 2.685 | −1.198 | 0.781 | A |
| Unaggressiveness | −1.522 | 1.141 | −1.185 | 0.067 | −1.505 | 1.273 | −1.301 | 0.569 | A |
| Intelligence | −1.819 | 2.799 | −1.915 | 2.687 | −2.009 | 3.084 | −1.883 | 3.112 | K |
| Interaction with tourists | −1.892 | 2.813 | −1.996 | 3.028 | −2.125 | 3.828 | −2.545 | 6.541 | K |
| A free lifestyle | −1.46 | 0.684 | −1.186 | −0.217 | −1.512 | 1.005 | −1.628 | 1.372 | A |
| Chinese traditional culture | −2.639 | 7.272 | −3.887 | 21.313 | −2.044 | 3.943 | −2.214 | 4.738 | K |
| Pet ownership | −2.311 | 5.27 | −3.152 | 14.619 | −2.379 | 5.738 | −2.123 | 4.504 | K |
| Donation | −3.14 | 10.459 | −3.297 | 14.607 | −2.498 | 6.12 | −2.341 | 5.707 | K |
| Willingness to learn | −3.102 | 9.966 | −4.334 | 26.831 | −3.118 | 9.432 | −1.755 | 2.074 | K |
| Knowledge about the animal | −3.609 | 14.157 | −4.072 | 21.848 | −2.816 | 7.985 | −2.297 | 5.357 | K |
| Preference | −2.587 | 6.764 | −4.558 | 27.698 | −2.443 | 5.719 | −2.497 | 6.613 | K |

Note. * A means ANOVA and K refers to Kruskal-Wallis.

## 4. Results

Descriptive analysis illustrates that 62.2% of the 659 participants (*n* = 410) were women, and 37.8% were men (*n* = 249). A total of 151 respondents were from first-tier cities, 338 were from second-tier cities, and 170 reported other origins. Ages in the dataset ranged from 7 to 80 years, with a mean of 30.31. In addition, 58.9% of participants (*n* = 388) had a bachelor's or a postgraduate degree. Younger visitors and women were more willing to participate in the survey than older generations and males. It is noted that the use of QR codes demanded Internet literacy from participants, and older tourists often declined the survey invitation, stating that they knew little about mobile phones or that their eyesight was poor. The sample shows that 60.2% of participants (*n* = 397) came to the Panda Base for holidays, while 25.3% (*n* = 167) visited to socialize with friends or family. Just over one-tenth of participants (*n* = 59, 10.7%) visited the Panda Base seeking scientific or conservation education. Participants visiting for business (*n* = 12, 2.2%) and other purposes (*n* = 27, 4.9%) were far less abundant.

Table 5 shows how the four groups of visitors perceived their preference for the four animals at Panda Base. One clear message is that the giant panda remained the most preferred animal across the multidimensional scale items. The red panda was the second preferred animal, followed by the peafowl and swan.

For all variables, the panda was deemed superior to the other three animals, with only a slight difference when first-time visitors with panda encounters perceived the unaggressiveness of the giant panda (m = 2.06).

For the giant pandas, the four groups of visitors showed significant differences in the animal's perceived cuteness (means between 1.05–1.19, *p* = 0.012), unaggressiveness (means between 1.69–2.06, *p* = 0.019), desired interaction with tourists (means between 1.31–1.53, *p* = 0.050), and pet ownership (means between 1.15–1.32, *p* = 0.043). Figure 1 illustrates the fluctuations of the four measures within the four visitor groups. Repeat visitors who had encountered pandas (Group 2b rated the panda cutest (m = 1.05) of all visitor groups. Notably, Group 2a and Group 2b evaluated the pandas as cuter than visitors without animal experience. Repeat visitors who had not yet seen the pandas (Group 1b) considered the giant panda to be the least aggressive (m = 1.69), though interaction with the animal can change this perception significantly (m = 1.94). An encounter with the giant panda also helped visitors rate their interactions higher. In particular, repeat visitors with panda encounters believed they enjoyed the best interaction with the animal (m = 1.31). Group 2a were most likely to claim their interest in owning a giant panda as a pet (m = 1.15), whereas repeat visitors were less likely to do so.

The red panda was considered the second most preferred and favored animal, and the animal (m = 2.00) was considered less aggressive than the giant panda (m = 2.06) by first-time visitors with animal encounters (Group 2a). Figure 2 shows how encounters with animals altered visitors' perceptions of the red panda. Red panda encounters were perceived to be less cute in comparison to the giant panda. First-time visitors seeing red pandas (Group 2a) believed the animal was less cute (m = 2.40) in the four visitor groups and created a more significant gap with visitors yet to see animals (Group 1a). Repeat visitors also tended to view the red panda as less cute after seeing the animal (m = 2.19) than repeat visitors without animal encounters (m = 2.05).

**Table 5.** Multidimensional animal preference scale measures in 4 visitor groups.

| Measures # | Group 1a | | Group 1b | | Group 2a | | Group 2b | | Total | | ANOVA | | Kruskal-Wallis |
|---|---|---|---|---|---|---|---|---|---|---|---|---|---|
| | m | SD | m | SD | m | SD | m | SD | m | SD | F | p | p |
| Giant panda | | | | | | | | | | | | | |
| Cuteness * | 1.19 | 0.489 | 1.1 | 0.433 | 1.16 | 0.467 | 1.05 | 0.219 | 1.13 | 0.421 | 3.673 | 0.012 * | 0.002 *^ |
| Unaggressiveness * | 1.82 | 1.119 | 2.06 | 1.211 | 1.69 | 1.065 | 1.94 | 1.171 | 1.88 | 1.148 | 3.328 | 0.019 * | 0.016 * |
| Intelligence | 1.43 | 0.866 | 1.43 | 0.882 | 1.43 | 0.836 | 1.34 | 0.778 | 1.41 | 0.841 | 0.444 | 0.722 | 0.612 |
| Interaction with tourists * | 1.53 | 0.947 | 1.4 | 0.844 | 1.53 | 0.958 | 1.31 | 0.772 | 1.44 | 0.888 | 2.325 | 0.074 | 0.050 * |
| A free lifestyle | 1.7 | 1.119 | 1.57 | 0.994 | 1.64 | 1.088 | 1.46 | 0.992 | 1.59 | 1.052 | 1.627 | 0.182 | 0.062 |
| Chinese traditional culture | 1.13 | 0.504 | 1.07 | 0.336 | 1.08 | 0.362 | 1.06 | 0.35 | 1.09 | 0.395 | 0.869 | 0.457 | 0.708 |
| Pet ownership * | 1.25 | 0.681 | 1.15 | 0.474 | 1.32 | 0.676 | 1.26 | 0.67 | 1.25 | 0.634 | 2.135 | 0.095 | 0.043 * |
| Donation | 1.15 | 0.501 | 1.13 | 0.583 | 1.15 | 0.519 | 1.1 | 0.376 | 1.13 | 0.501 | 0.373 | 0.772 | 0.445 |
| Willingness to learn | 1.15 | 0.489 | 1.11 | 0.458 | 1.11 | 0.369 | 1.09 | 0.386 | 1.12 | 0.428 | 0.599 | 0.616 | 0.475 |
| Knowledge about the animal | 1.18 | 0.565 | 1.1 | 0.466 | 1.14 | 0.515 | 1.07 | 0.321 | 1.12 | 0.478 | 1.767 | 0.152 | 0.067 |
| Preference | 1.17 | 0.489 | 1.1 | 0.404 | 1.13 | 0.387 | 1.11 | 0.49 | 1.13 | 0.445 | 0.867 | 0.458 | 0.106 |
| Red panda | | | | | | | | | | | | | |
| Cuteness * | 2.1 | 0.686 | 2.4 | 0.798 | 2.05 | 0.6 | 2.19 | 0.608 | 2.19 | 0.689 | 8.839 | 0.000 * | 0.000 * |
| Unaggressiveness | 1.99 | 0.823 | 2 | 0.896 | 2.12 | 0.815 | 1.98 | 0.783 | 2.02 | 0.83 | 0.897 | 0.398 | 0.343 |
| Intelligence | 2.04 | 0.66 | 2.03 | 0.633 | 2.02 | 0.602 | 2.02 | 0.621 | 2.03 | 0.628 | 0.053 | 0.984 | 0.957 |
| Interaction with tourists | 2.04 | 0.745 | 2.17 | 0.725 | 2.18 | 0.751 | 2.06 | 0.592 | 2.11 | 0.709 | 1.678 | 0.17 | 0.156 |
| A free lifestyle | 2.1 | 0.731 | 2.25 | 0.746 | 2.13 | 0.651 | 2.16 | 0.572 | 2.16 | 0.68 | 1.388 | 0.245 | 0.298 |
| Chinese traditional culture | 2.29 | 0.639 | 2.35 | 0.821 | 2.36 | 0.728 | 2.24 | 0.621 | 2.31 | 0.707 | 1.065 | 0.363 | 0.33 |
| Pet ownership | 1.98 | 0.547 | 2.09 | 0.735 | 1.96 | 0.672 | 2 | 0.528 | 2.01 | 0.627 | 1.401 | 0.241 | 0.092 |
| Donation | 2.04 | 0.54 | 2.15 | 0.669 | 2.08 | 0.561 | 2.05 | 0.501 | 2.08 | 0.571 | 1.15 | 0.328 | 0.178 |
| Willingness to learn | 2.04 | 0.494 | 2.05 | 0.636 | 2.07 | 0.552 | 2.06 | 0.453 | 2.06 | 0.537 | 0.091 | 0.965 | 0.985 |
| Knowledge about the animal | 2.04 | 0.517 | 2.11 | 0.648 | 2.14 | 0.61 | 2.16 | 0.569 | 2.11 | 0.588 | 1.262 | 0.286 | 0.279 |
| Preference | 2.02 | 0.547 | 2.09 | 0.623 | 2.07 | 0.613 | 2.03 | 0.412 | 2.05 | 0.556 | 0.483 | 0.694 | 0.45 |
| Peafowl | | | | | | | | | | | | | |
| Cuteness | 3.07 | 0.643 | 3.09 | 0.67 | 3.13 | 0.583 | 3.24 | 0.6 | 3.13 | 0.627 | 2.373 | 0.069 | 0.082 |
| Unaggressiveness | 2.76 | 0.754 | 2.64 | 0.8 | 2.69 | 0.845 | 2.64 | 0.917 | 2.68 | 0.83 | 0.773 | 0.509 | 0.591 |
| Intelligence | 2.96 | 0.754 | 2.99 | 0.633 | 2.98 | 0.698 | 3.04 | 0.65 | 2.99 | 0.683 | 0.472 | 0.702 | 0.704 |
| Interaction with tourists * | 2.76 | 0.754 | 2.83 | 0.717 | 2.67 | 0.791 | 2.93 | 0.638 | 2.8 | 0.733 | 3.657 | 0.012 * | 0.020 * |
| A free lifestyle | 2.77 | 0.732 | 2.91 | 0.723 | 2.79 | 0.83 | 2.92 | 0.72 | 2.85 | 0.755 | 1.723 | 0.161 | 0.185 |
| Chinese traditional culture * | 2.83 | 0.625 | 2.79 | 0.643 | 2.89 | 0.612 | 3 | 0.54 | 2.88 | 0.61 | 3.567 | 0.014 * | 0.014 * |
| Pet ownership | 3.03 | 0.518 | 3.06 | 0.635 | 3.03 | 0.55 | 3.08 | 0.632 | 3.05 | 0.584 | 0.255 | 0.858 | 0.562 |
| Donation * | 3 | 0.451 | 2.98 | 0.598 | 3.03 | 0.493 | 3.13 | 0.465 | 3.03 | 0.507 | 2.988 | 0.031 * | 0.033 * |
| Willingness to learn | 3.02 | 0.495 | 3.02 | 0.572 | 3.01 | 0.408 | 3.15 | 0.542 | 3.05 | 0.509 | 2.857 | 0.036 * | 0.066 |
| Knowledge about the animal | 2.96 | 0.488 | 2.96 | 0.565 | 2.94 | 0.496 | 3.03 | 0.533 | 2.97 | 0.521 | 0.915 | 0.433 | 0.36 |
| Preference | 3.05 | 0.516 | 2.96 | 0.597 | 3.04 | 0.48 | 3.09 | 0.469 | 3.04 | 0.519 | 1.679 | 0.17 | 0.214 |

**Table 5.** *Cont.*

| Measures # | Group 1a | | Group 1b | | Group 2a | | Group 2b | | Total | | ANOVA | | Kruskal-Wallis |
|---|---|---|---|---|---|---|---|---|---|---|---|---|---|
| | m | SD | m | SD | m | SD | m | SD | m | SD | F | p | p |
| Swan | | | | | | | | | | | | | |
| Cuteness * | 3.64 | 0.612 | 3.41 | 0.768 | 3.66 | 0.598 | 3.52 | 0.664 | 3.56 | 0.669 | 4.994 | 0.002 * | 0.004 * |
| Unaggressiveness | 3.43 | 0.926 | 3.3 | 1.007 | 3.5 | 0.81 | 3.44 | 0.846 | 3.42 | 0.901 | 1.475 | 0.22 | 0.402 |
| Intelligence | 3.57 | 0.747 | 3.55 | 0.855 | 3.58 | 0.828 | 3.6 | 0.721 | 3.57 | 0.789 | 0.119 | 0.949 | 0.901 |
| Interaction with tourists | 3.67 | 0.633 | 3.6 | 0.806 | 3.62 | 0.772 | 3.69 | 0.675 | 3.64 | 0.724 | 0.653 | 0.581 | 0.864 |
| A free lifestyle | 3.42 | 0.988 | 3.27 | 1.123 | 3.43 | 0.95 | 3.46 | 0.946 | 3.4 | 1.004 | 1.178 | 0.317 | 0.556 |
| Chinese traditional culture | 3.76 | 0.574 | 3.7 | 0.755 | 3.67 | 0.632 | 3.7 | 0.624 | 3.71 | 0.648 | 0.477 | 0.699 | 0.503 |
| Pet ownership | 3.74 | 0.558 | 3.61 | 0.812 | 3.69 | 0.656 | 3.66 | 0.654 | 3.68 | 0.676 | 1.171 | 0.32 | 0.492 |
| Donation | 3.8 | 0.539 | 3.65 | 0.828 | 3.74 | 0.6 | 3.72 | 0.597 | 3.73 | 0.65 | 1.568 | 0.196 | 0.287 |
| Willingness to learn * | 3.79 | 0.581 | 3.73 | 0.712 | 3.8 | 0.57 | 3.7 | 0.571 | 3.76 | 0.611 | 1.065 | 0.363 | 0.042 * |
| Knowledge about the animal | 3.81 | 0.558 | 3.73 | 0.762 | 3.78 | 0.575 | 3.74 | 0.554 | 3.76 | 0.617 | 0.553 | 0.647 | 0.42 |
| Preference | 3.75 | 0.587 | 3.76 | 0.719 | 3.76 | 0.562 | 3.77 | 0.53 | 3.76 | 0.602 | 0.023 | 0.995 | 0.799 |

Note: The bigger m suggests that the animal took a lower rank. * indicates $p < 0.05$. # Means (m) of the data were calculated based on Table 3. The scale indicates that when the number is bigger, the animal received less preference in this dimension. ˆ The highlighted level of significance in ANOVA and Kruskal–Wallis indicates the $p$ selected based on the normality of the samples.

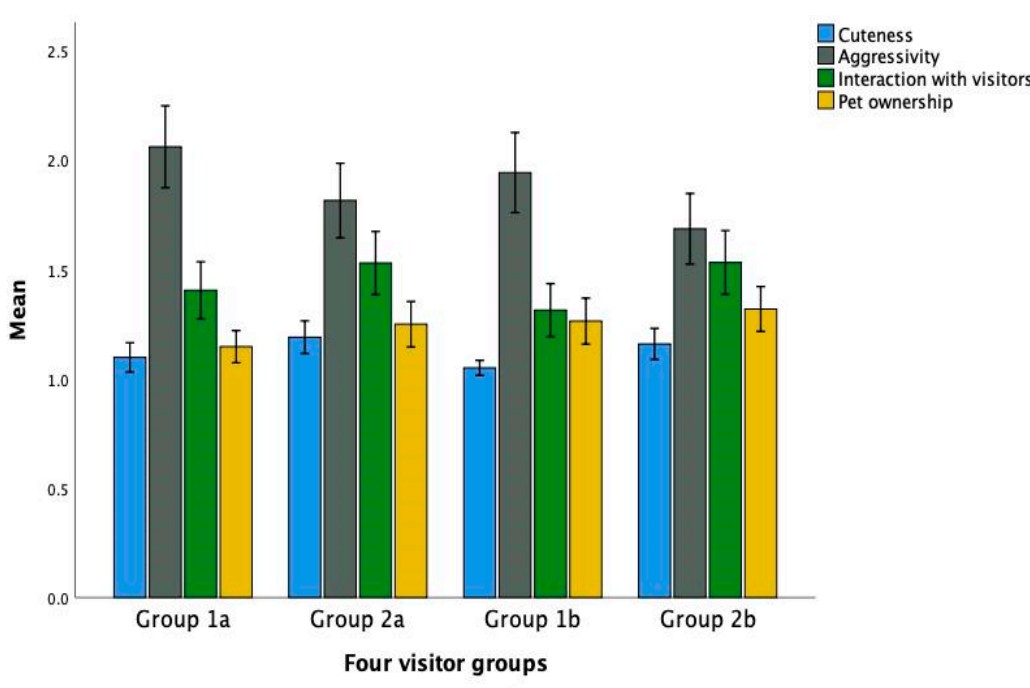

**Figure 1.** Giant panda multidimensional measures with significant differences between the four groups.

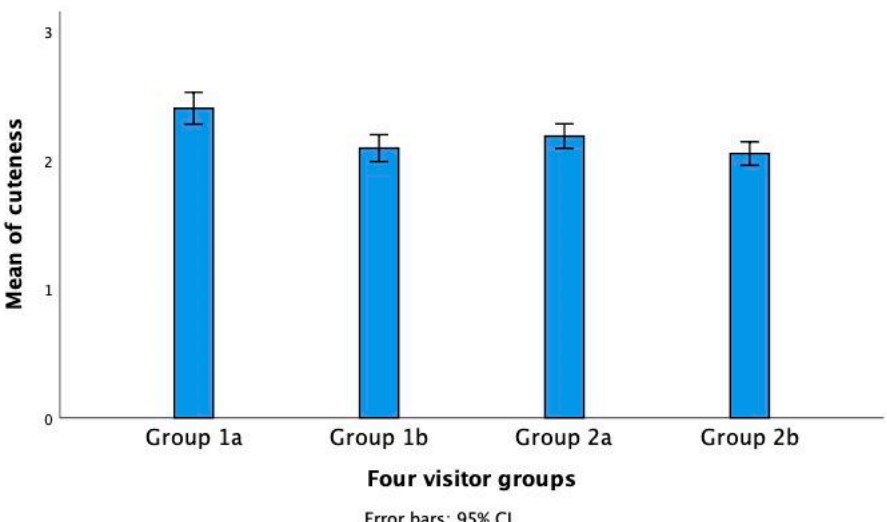

**Figure 2.** Red panda multidimensional measures with significant differences between the four groups.

The perception of the peafowl fluctuated more frequently for the four groups of visitors. Perceptions concerning the peafowl's cuteness ($p = 0.069$), interaction with tourists ($p = 0.012$), relation to Chinese traditional culture ($p = 0.014$), donation ($p = 0.031$), and willingness to learn ($p = 0.036$) fluctuated significantly between groups of visitors even though the peafowls maintained their place as the third preferred animal among the four.

Figure 3 shows that peafowls were considered more interactive (m = 2.80) than cute (m = 3.13). An overall trend in Figure 3 is that visitor experiences and animal encounters lowered the values of peafowls on all dimensions. A notable fluctuation was found in the animal's interaction with visitors. Repeat visitors without animal encounters (Group 2b) tended to believe the animal was the most interactive (m = 2.67), but they also dismissed the idea quickly after encountering the animal (m = 2.93). First-time visitors who met the animal (Group 2a) tended to highlight peafowls' link with traditional Chinese culture (m = 2.79) more than other groups. Repeat visitors tended to be less concerned about the animal's cuteness and willingness to donate to and learn about the animal than first-time visitors.

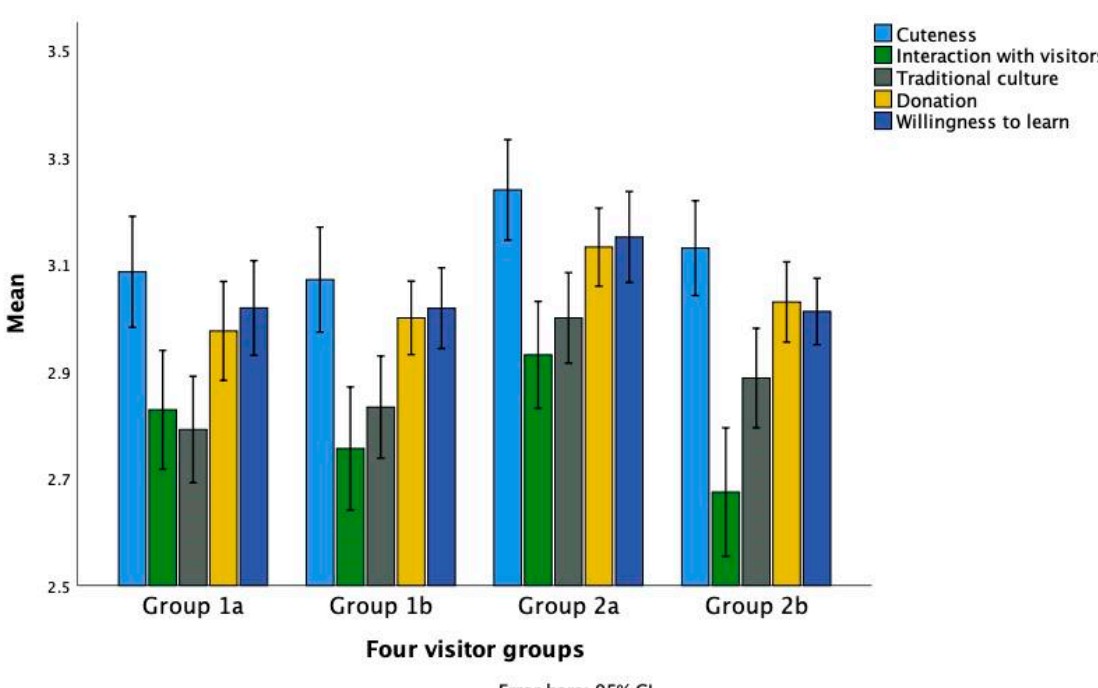

**Figure 3.** Peafowl multidimensional measures with significant differences between the four groups.

The swan was considered the least favorite animal, as Table 4 indicates, with significant differences within the four groups of visitors in terms of cuteness ($p = 0.002$) and visitors' willingness to learn about the animal. Figure 4 shows that, in contrast to red pandas, animal encounters added to the perceived level of cuteness and willingness to learn about the swans for both first-time visitors (Group a) and repeat visitors (Group b). Notably, the peafowl's cuteness was not a significant factor attracting people to revisit Panda Base (Group 1a).

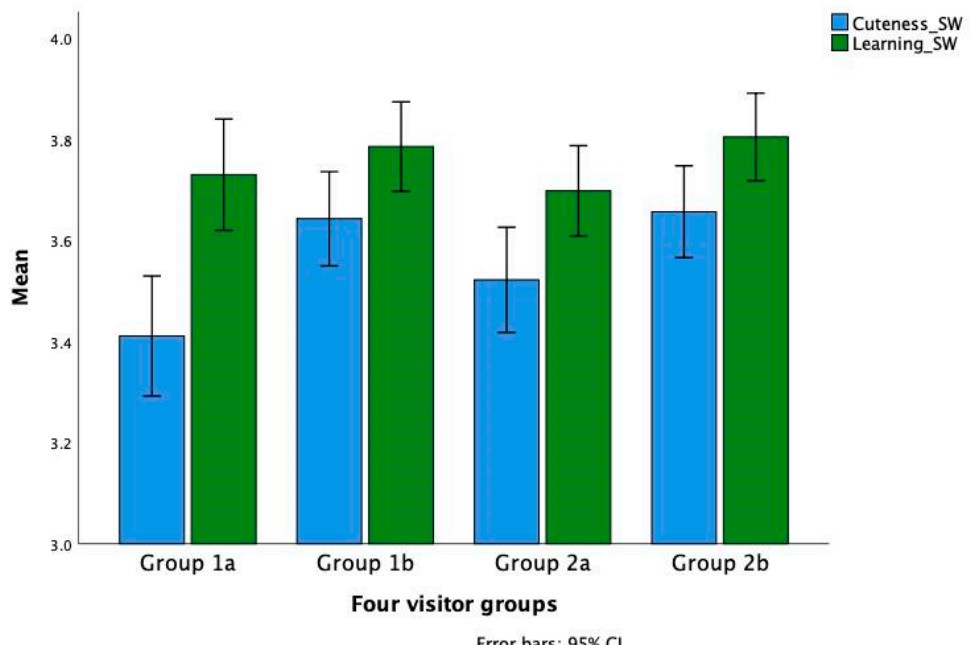

**Figure 4.** Swan multidimensional measures with significant differences between the four groups.

## 5. Discussion

The findings of this study reveal that lively capital and encounter value are dynamic outcomes of a learning process centered around visitors and animals at the Panda Base. By examining the multidimensional perceptions of animals among four groups of visitors, the results demonstrate that affection and preference for animals are not static and unchanging over time. The preference for a particular species can be understood as a learning process that continues to revolve around visitor–animal interactions. Furthermore, different animals have varying abilities to influence public preferences and emotions, meaning that visitors can generate varied outcomes as the learning process differs. The public's preference towards animals can also result from learning and acquirement rather than an innate desire or instinct. Lively capital and encounter value through the lens of consumer learning provide valuable insights into this complex interplay. Viewing visitors' preference for animals through the lens of learning also suggests that conservation education can play an active rather than responsive role in shaping and forming the public's interest in animals.

The primary message of this study underscores the dynamic nature of lively capital, involving the accumulation, creation, and circulation of value through animal–visitor interactions. Our results show that visitor preferences for four animals showcased at the Panda Base evolved across various dimensions, influenced by the specific animals encountered and visitation frequency. As a learning process, visitor preferences for the giant panda, red panda, peafowl, and swan exhibited distinct trajectories—encounters with these animals could either enhance or diminish the level of affection and preference for them. At times, encounters had contrasting effects on lively capital. For instance, while interactions with animals increased the preference for the giant panda and swans, encounters with the red panda and peafowl diminished the appeal of these animals.

Additionally, the study observed dynamics within the lively capital generated by a single species. For example, the giant panda was perceived as cuter and less aggressive after encounters. As a learning journey, preferences and emotions towards animals depend on a complex interplay between species, leading to diverse outcomes in the consideration of different animals [30].

While animal–visitor encounters provide learning opportunities, the researchers propose that lively capital is influenced by both short-term and long-term impacts resulting from these encounters. Short-term impacts are reflected in animal encounters between Groups A and B. In contrast, long-term impacts are seen in animal preferences between first-time and repeat visitors from Groups 1 and 2. Although this study highlights notable fluctuations between Groups A and B (Figures 1–4), the enduring influence, evident in the changing animal preferences among first-time and repeat visitors, adds another dimension to the accumulation of lively capital in wildlife tourism. It is observed that the perceived level of cuteness tended to increase over time for the giant panda ($p = 0.002$) and red panda ($p = 0.000$). In contrast, the peafowl experienced a significant decline in lively capital, and the swan remained relatively stable. While this study did not delve into the learning curve between first-time and repeat visitors, it is reasonable to assume that the giant panda's celebrity status [60,61] and the increasing visibility of pandas (red and giant) on social media [62,63] played a crucial role in boosting preferences and affection for these animals among repeat visitors. In contrast, peafowls and swans, two species receiving far less attention in social media, experienced a long-term decline in their lively capital. The study results show that visitation frequency can generate varying encounter values for different species.

The study proposes that learning driven by animal preference and affection leads to the accumulation of lively capital centered around the experience and culture of consumption. Unlike the red panda, peafowls, and swans, the giant panda is the sole animal among the four living in enclosures at Panda Base. Visitors sometimes had the opportunity for close encounters with peafowls, swans, and red pandas as they freely roamed the area. However, interactions with the giant panda surpassed visitors' initial expectations, leading to a more favorable evaluation of their interactions with this animal. Such a result indicates that the

giant panda was perceived as more interactive than visitors had anticipated. Conversely, all free-roaming animals were considered less interactive after visitors' encounters.

Why did visitors view the enclosed giant panda as being more interactive than the free-roaming animals? As one of the planet's most anthropomorphized animals, the giant panda amalgamates visitors' access to anthropocentric knowledge, their motivation to understand the animal, and their desire for affiliation [64]. These factors collectively contribute to visitors' ability to connect and form bonds with the giant panda. We argue that the lack of anthropomorphic attributes associated with the other three animals diminished their perceived interactive value despite visitors having direct access to these animals. In this context, visitors possessed more knowledge about giant pandas, enabling them to establish richer interactions with these animals.

Additionally, the study highlights social media and the Internet's pivotal role in the long-term accumulation of lively capital acting as influential agents shaping human–animal relations, becoming focal areas for conservation programs. As highlighted in Lupton's recent book [63], digital technologies and datafication have been instrumental in transforming human–animal relations. Our study underscores the impact of social media on the sustained accrual of lively capital. Consequently, conservation programs must extend their scope beyond mere human–animal encounters, emphasizing how individuals maintain connections with animals in their daily lives. Given the varying degrees of commodification among different animal species in lively capital, understanding the proclivity for appreciation or depreciation of a species can aid, therefore, in the restructuring of conservation programs, often founded on animal charisma [18,19]. For instance, lesser-known animals at the Panda Base, such as the red panda, peafowl, and swan, could benefit from conservation initiatives by introducing these species to visitors through interpretive measures.

## 6. Conclusions

Recent studies demonstrate the value of lively capital as a critical framework for examining human–animal relationships. This study operationalized lively capital in wildlife tourism experiences and used empirical data to analyze visitors' interactions with animals and their visitation patterns. To gain insight into the practical value of lively capital, our study demonstrates that it is most effectively comprehended as a multifaceted and dynamic process rooted in visitors' experiential learning within a consumption-driven society. The study posits that the public's preferences and affections toward animals are not stagnant but rather part of a co-evolving process intrinsic to the creation of encounter value and the accumulation of lively capital.

This study further reveals that face-to-face interactions, encompassing activities, like touching, feeding, and intimate contact with animals, may not necessarily heighten visitors' perceived levels of interaction with animals. In the case of animals that are less anthropomorphized or integrated into human social life, these interactions may not hold the significance that managers and marketers in wildlife tourism have often assumed and promoted. Paradoxically, direct encounters with certain animals can potentially diminish their perceived interactivity among visitors. Conversely, encounters characterized by visually rich and value-laden experiences, such as those with enclosed giant pandas, tend to be regarded as more interactive by visitors. This discovery has profound implications for animal organizations and institutions, particularly those with visitor guidelines that aim to establish ethical boundaries for direct animal interactions. The researchers advocate for carefully aligning conservation programs with each species' unique, lively capital, offering guidance on whether to encourage or discourage further contact with animals.

**Supplementary Materials:** The following supporting information can be downloaded at: https://www.mdpi.com/article/10.3390/jzbg5010002/s1.

**Author Contributions:** Conceptualization, Y.G. and D.F.; methodology, Y.G.; formal analysis, Y.G.; investigation, Y.G.; resources, Y.G.; data curation, Y.G.; writing—original draft, Y.G.; writing—review

and editing, D.F.; visualization, Y.G.; supervision, D.F.; project administration, Y.G. All authors have read and agreed to the published version of the manuscript.

**Funding:** This research received no external funding.

**Institutional Review Board Statement:** The study was conducted in accordance with the Declaration of Helsinki. Ethical review and approval were waived for this study in accordance with the Personal Information Protection Law of the People's Republic of China. The law does not define anonymized information as personal information.

**Informed Consent Statement:** All subjects gave their informed consent for inclusion before they participated in the study.

**Data Availability Statement:** The data presented in this study are openly available in FigShare at https://doi.org/10.6084/m9.figshare.24657393.v1. Accessed on 22 December 2023.

**Acknowledgments:** We thank Qiaolin Chen for her assistance in data collection.

**Conflicts of Interest:** The authors declare no conflicts of interest.

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
