# Peer review of "Preference for Animals: A Comparison of First-Time and Repeat Visitors"

_2673-5636, doi:10.3390/jzbg5010002_

Round 1

Reviewer 1 Report

Comments and Suggestions for Authors

Dear Authors,

Thank you for submitting this manuscript that explores the perception of four species being visited by the public. This is a potentially interesting and useful manuscript in terms of public preference for animals.

At current however, there seem to be some large revisions required in the manuscript to ensure the work is scientifically robust. I have attached the PDF version of the manuscript with specific comments. Additionally, please consider the following points: 

1, Transparency. Currently, the questions being asked to the audiences are unclear. Sampling regimes for visitors are similarly unclear, and this makes the results difficult to interpret. Please provide a more in0depth explanation of the sampling methods used in your work. This is essential for repeatability.

2. ANOVA.  ANOVA is a parametric test and assumes normal distribution of data. Did you test your data for normality and if so what was the finding? Please report it in the work. If the data are not parametric an alternative test (such as Kruskal Wallis) would be more appropriate.

3. References and citations. No attempt has been made at preparing these to the style requirements of the journal. If the work is to be resubmitted, please ensure the instructions are followed carefully.

4. Formatting of figures and tables. Please ensure that error is incorporated into these (e.g. as range, error bars etc). Ensure the tables and figures are formatted as per the requirements of the journal and to a professional standard.

5. Wording. In several sections the writing is quite unscientific and there is frequent use of 'we'. It is often unclear who 'we' refers to. Additionally, there are several sweeping and unsubstantiated claims. Please ensure any claims made in the work are fully supported with citations or removed.

Comments on the Quality of English Language

Currently there are grammatical errors and the work has not been suitably formatted to the style requirements of the journals.

Reviewer 2 Report

Comments and Suggestions for Authors

This is a fantastic study and great paper. I look forward to citing it often!

The only 2 recommendations I have are to redo the figures. Need to add much more detail around means, error bars, variable labels etc. As is, they are insufficient to provide reader any meaningful information.

Also, in the results, need to add ANOVA and t-test results to discuss differences between means across groups. Be sure to refer to statistic as "mean" and not "average". Add info to tables where appropriate to highlight diffs between means, p values, etc.

Round 2

Reviewer 1 Report

Comments and Suggestions for Authors

Dear Authors,

Thank you for submitting a revised version of this manuscript. It appear that the reviewer points have been carefully addressed, and it was good to see the inclusion of the questionnaire. While the edits have resulted in a more repeatable study, there remain some major errors in citation formatting and reference style. Please ensure these are addressed in full. Please also ensure a full proof read as there appear to be incomplete sentences present for example in the introduction still.

Comments on the Quality of English Language

Major improvements have been made. However, there remain some incomplete sentences and grammatical inaccuracies (e.g. see introduction) that need to be addressed.
